# Inhibition of Aldose Reductase by Ginsenoside Derivatives via a Specific Structure Activity Relationship with Kinetics Mechanism and Molecular Docking Study

**DOI:** 10.3390/molecules27072134

**Published:** 2022-03-25

**Authors:** Md Yousof Ali, Sumera Zaib, Susoma Jannat, Imtiaz Khan, M. Mizanur Rahman, Seong Kyu Park, Mun Seog Chang

**Affiliations:** 1Department of Physiology and Pharmacology, Hotchkiss Brain Institute and Alberta Children’s Hospital Research Institute, Cumming School of Medicine, University of Calgary, Calgary, AB T2N 4N1, Canada; mdyousof.ali@ucalgary.ca; 2Department of Biochemistry, Faculty of Life Sciences, University of Central Punjab, Lahore 54590, Pakistan; sumera.biochem@gmail.com; 3Department of Biochemistry and Molecular Biology, University of Calgary, Calgary, AB T2N 1N4, Canada; jannatacct@gmail.com; 4Department of Chemistry, Manchester Institute of Biotechnology, The University of Manchester, 131 Princess Street, Manchester M1 7DN, UK; imtiaz.khan@manchester.ac.uk or; 5Department of Biotechnology and Genetic Engineering, Faculty of Biological Science, Islamic University, Kushtia 7003, Bangladesh; mmrahmanbtg79@hotmail.com; 6Department of Prescriptionology, College of Korean Medicine, Kyung Hee University, 26 Kyunghee dae-ro, Dongdaemun-gu, Seoul 02447, Korea; comskp@khu.ac.kr; 7Qgenetics, Seoul Bio Cooperation Center 504, 23 Kyunghee dae-ro, Dongdaemun-gu, Seoul 02447, Korea

**Keywords:** ginsenosides, diabetic complication, aldose reductase, enzyme kinetics, molecular docking, sorbitol accumulation

## Abstract

This present work is designed to evaluate the anti-diabetic potential of 22 ginsenosides via the inhibition against rat lens aldose reductase (RLAR), and human recombinant aldose reductase (HRAR), using _DL_-glyceraldehyde as a substrate. Among the ginsenosides tested, ginsenoside Rh2, (20*S*) ginsenoside Rg3, (20*R*) ginsenoside Rg3, and ginsenoside Rh1 inhibited RLAR significantly, with IC_50_ values of 0.67, 1.25, 4.28, and 7.28 µM, respectively. Moreover, protopanaxadiol, protopanaxatriol, compound K, and ginsenoside Rh1 were potent inhibitors of HRAR, with IC_50_ values of 0.36, 1.43, 2.23, and 4.66 µM, respectively. The relationship of structure–activity exposed that the existence of hydroxyl groups, linkages, and their stereo-structure, as well as the sugar moieties of the ginsenoside skeleton, represented a significant role in the inhibition of HRAR and RLAR. Additional, various modes of ginsenoside inhibition and molecular docking simulation indicated negative binding energies. It was also indicated that it has a strong capacity and high affinity to bind the active sites of enzymes. Further, active ginsenosides suppressed sorbitol accumulation in rat lenses under high-glucose conditions, demonstrating their potential to prevent sorbitol accumulation ex vivo. The findings of the present study suggest the potential of ginsenoside derivatives for use in the development of therapeutic or preventive agents for diabetic complications.

## 1. Introduction

According to the World Health Organization, approximately 200 million people—5.9% of the adult population worldwide—suffer from diabetes, a very common disease that has assumed epidemic proportions. As a result, it is expected that by 2025, the disease will have a significant impact on human health [1,2]. Diabetes mellitus (DM) is a complex and chronic metabolic disease characterized by hyperglycemia, a condition in which a significant amount of glucose enters in the polyol pathway, which is considered as the main cause of the pathogenesis of long-term diabetic complications. Aldose reductase (AR) is a major enzyme in the polyol pathway that is implicated in the development of diabetes complications, such as neuropathy, nephropathy, and retinopathy, as well as cataract formation [3,4,5,6,7]. In the diabetic state, AR also plays a critical role in hyperglycemia-stimulated oxidative stress, which results in tissue and vascular damage [8]. AR, a nicotinamide adenine dinucleotide phosphate (NADPH)-dependent oxido-reductase, is an important enzyme that catalyzes the reduction of glucose to sorbitol, which sorbitol dehydrogenase metabolizes further to fructose [9]. Many mechanistic models have been proposed to offset these complications and among them, the polyol pathway is accepted widely as a potential target to address diabetic complications [10]. The identification of novel AR inhibitors from natural sources with few side effects would be of great clinical significance in the management of diabetic complications, and most herbal medicines are nontoxic and have few side effects [11,12,13,14].

Ginseng belongs to the Araliaceae family (genus Panax). It is a highly cherished perennial herb, as well as a common medicinal plant. It has been applied for thousands of years as a folk herbal medicine in Asia, mainly China, Korea, and Japan. So far, 14 (fourteen) plants have been classified within the genus Panax, including 12 (twelve) species and 2 (two) infraspecific taxa [15]. *P. notoginseng* (Burk.), *P. quinquefolius* L, and *P. ginseng* (Meyer) are the three main commercial varieties [16]. *P. ginseng* is the most well known in East Asian. However, presently it is cultivated in worldwide, such as Russia, Japan, Canada, and the United States [17]. Ginseng has different affects, such as increasing energy and strengthening the spleen. It benefits the lungs, heart, and mind, and nourishes fluid [18,19,20]. Moreover, Korean red ginseng is recognized to have several biological activities, such as improving the strengthening ability of the immune system, blood circulation, memory, fatigue and menopause, and antioxidants [21,22,23]. Previously, ginseng was reported to be effective in preventing and treating diabetic complications, such as nephropathy [24,25].

The beneficial effects of ginseng are attributable principally to ginsenosides, which are the main active ingredients. Currently, close to about 150 (one hundred fifty) ginsenosides have been recognized. Their basic structure is slightly similar with 4 (four) steroid nuclei rings and 30 (thirty) carbon atoms [19]. Three types of ginsenosides are found and they are distinguished according to the number and position of sugar, namely the panaxatriol, panaxadiol, and oleanolic groups [19]. It was reported that ginsenosides have plentiful biological activities, such as antioxidant, anti-inflammatory, anti-cancer, antifatigue, antiaging, anti-Alzheimer’s disease, anti-diabetic activities, and physiological and neuroprotective functions [26,27,28,29,30,31,32]. Recently, we reported that ginsenoside derivatives have promising antiglycation properties that may be able to prevent the progression of diabetic complications [33]. In other studies, we reported that ginsenosides have potential antihypertensive effects via the inhibition of the angiotensin converting enzyme (ACE), as well as antioxidant properties [34].

Despite the promising biological potential of ginseng extract and its major active ingredients, ginsenosides, no systematic studies have been conducted on the RLAR and HRAR inhibitory activities of ginsenoside derivatives. Therefore, as part of our continuing efforts to identify potent AR agents derived from natural sources, we explored the antidiabetic potential of these derivatives. Here, we also conducted the enzyme kinetic analyses through Lineweaver-Burk plots. RLAR and HRAR were used for the molecular docking analysis. HRAR and RLAR inhibition mechanisms and docking energies were tested. In addition, we also examined the effect of active ginsenosides on sorbitol accumulation to evaluate their potential to treat diabetic complications. We also addressed the structure–activity relationship of ginsenoside derivatives, which may be used to prevent diabetes complications.

## 2. Results and Discussion

### 2.1. RLAR Inhibitory Activities of Ginsenoside Derivatives and Analysis of Structure-Activity Relation

To evaluate the potential activity of the ginsenosides against diabetic complications, we examined whether they inhibited RLAR, and as Table 1 shows, they did so significantly. The most potent RLAR inhibitors were ginsenoside Rh2, (20*S*) ginsenoside Rg3, (20*R*) ginsenoside Rg3, and ginsenoside Rh1, with IC_50_ values of 0.67 ± 0.01, 1.25 ± 0.28, 4.28 ± 0.31, and 7.28 ± 0.27 µM, respectively, relative to the positive control quercetin (4.88 ± 0.71 µM). In addition, ginsenoside Rf, (20*S*) ginsenoside Rg2, protopanaxadiol, protopanaxatriol, (20*R*) ginsenoside Rg2, compound K, ginsenoside Rg1, and ginsenoside Rg5 demonstrated significant inhibitory activities against RLAR, with IC_50_ values of 11.29 ± 1.49, 14.38 ± 0.99, 21.38 ± 2.45, 27.88 ± 1.19, 29.38 ± 2.33, 41.48 ± 3.99, 49.48 ± 1.88, and 54.47 ± 1.22 µM, respectively. Ginsenosides Rb1, Rb2, Rb3, Rd, Ra2, Rs1, Rs2, and Re demonstrated moderate RLAR inhibitory activity with IC_50_ values of 103.11 ± 7.45, 121.12 ± 3.43, 117.43 ± 6.89, 89.58 ± 2.99, 178.39 ± 7.39, 142.77 ± 3.77, 149.34 ± 4.22, and 81.27 ± 2.18 µM, respectively. Ginsenosides Rc and Ra1 were found to be inactive at the concentrations tested.

Structurally, the ginsenosides are characterized within the protopanaxatriol (PPT) or protopanaxadiol (PPD) groups (Figure 1). The inhibitory effects on RLAR were examined between the ginsenosides and the target enzyme inhibition, which are listed as in Table 1. The ginsenosides’ RLAR inhibitory activity is associated to multiple factors, such as the position of the hydroxyl groups, the number of sugar molecules, and the stereoselectivity. The different positions of the ginsenoside hydroxyl groups influenced the RLAR inhibition. The number of hydroxyl groups at the C-20 position without or with sugar fractions increased the ginsenosides’ RLAR inhibition. The ginsenoside RLAR inhibitory activity of Rh1, Rh2, Rf, Rg2, (20*R*) Rg2, (20*R*) Rg3, (20*S*) Rg3, (20*S*), protopanaxatriol, and protopanaxadiol were likened scientifically. Ginsenoside Rh1 and Rh2 comprise a single hydroxyl group at the C-20 location with a single sugar moiety at the C-3 or C-6 location. This result exhibited the utmost effective RLAR inhibitory action, whereas two or more sugar moieties at the C-6 or C-3 location decreased the RLAR inhibitory action of ginsenoside Rf, (20*S*) Rg2, (20*R*) Rg3, (20*S*) Rg3, and (20*R*) Rg2. Similarly, protopanaxatriol and aglycone protopanaxadiol comprise a hydroxyl group at the C-20 location that are engaged in RLAR inhibitory action. Later, the presence of a hydroxyl group at the C-20 location with a single sugar moiety on the ginsenoside molecule plays a dynamic role in inhibiting RLAR and declines the action of two or more sugar moieties. Previously, Fatmawati et al. [35] reported that the hydroxyl group at the C-20 position of ginsenosides plays an important role in aldose reductase inhibition, which is similar to our finding.

The RLAR inhibitory action enhanced the number of sugar moieties, whereas ginsenoside molecule reduced the number. Ginsenosides with three or more sugar molecules, such as Rb1, Rc, Rb2, Ra1, Rb3, Ra2, Rs2, Rs1, Re, and Rd, exhibited weak or no RLAR inhibitory action as compared to ginsenosides Rf, Rh2, Rh1, (20*S*) Rg3, (20*R*) Rg3, (20*S*) Rg2, (20*R*) Rg2. Ginsenosides Rf, Rh1, Rh2, (20*R*) Rg3, (20*S*) Rg2, (20*S*) Rg3, (20*R*) Rg2, PPT, PPD, and compound K displayed the solid RLAR inhibition, demonstrating that a partial number of sugar moieties at the C-3, C-6, or C-20 location possibly plays a significant role in ginsenoside action, and the exchange of more sugar residues enhances the decreased action.

Recently, Zhang et al. [36] reported that a traditional Chinese herbal extract (referred to as the Shenqi Jiangtang granule), which contains ginseng stem and leaf extract, demonstrated aldose reductase inhibitory activity. The authors also identified several ginsenoside derivatives and tested them against aldose reductase using rabbit crystalline lenses. According to the authors’ reports, the ginsenoside derivatives demonstrated significant inhibitory activity against aldose reductase, which differs from our data slightly, because we used a different source of lenses (rat lenses) and Zhang et al. [36] used rabbit crystalline lenses. The pure rat lens homogenate extract was used to test the AR activity of ginsenoside derivatives in the current study. However, there are some drawbacks to using rat lens crude homogenate because it contains other NADPH-dependent reductase enzymes that could affect specific AR activity [10]. As a result, for testing the AR activity of ginsenoside derivatives, we used a more selective pure grade HRAR enzyme.

### 2.2. HRAR Inhibitory Activities of Ginsenoside Derivatives and Analysis of Structure-Activity Relation

To evaluate the anti-diabetic activity of 22 ginsenoside derivatives, the inhibitory potential of human recombinant aldose reductase (HRAR) was evaluated using _DL_-glyceraldehyde as the substrate, and the results are expressed as IC_50_ values and presented in Table 1. Most of the tested compounds clearly demonstrated strong HRAR inhibitory activity. Particularly, protopanaxadiol, protopanaxatriol, and compound K exhibited the most potent HRAR inhibitory potential with IC_50_ values of 0.36 ± 0.1, 1.43 ± 0.14 and 2.23 ± 0.54 µM compared to the positive controls, quercetin and zenaresta, with IC_50_ values of 3.11 ± 0.22 and 0.69 ± 0.11µM, respectively. Ginsenoside Rh1, ginsenoside Rh2, (20*R*) ginsenoside Rg3, (20*S*) ginsenoside Rg3, (20*R*) ginsenoside Rg2, (20*S*) ginsenoside Rg2, ginsenoside Rf, and ginsenoside Rg1 displayed significant HRAR inhibitory activity with IC_50_ values of 4.66 ± 0.34, 7.44 ± 0.55, 8.67 ± 0.87, 9.92 ± 0.56, 13.66 ± 0.99, 15.67 ± 1.05, 19.45 ± 1.55, and 27.56 ± 2.12 µM, respectively. In comparison, ginsenosides Rd, Rg5, Re, Rc, Ra1, Rb3, Ra2, and Rb1 demonstrated moderate HRAR inhibitory activity, with IC_50_ values of 37.45 ± 1.33, 38.56 ± 2.91, 43.45 ± 3.11, 56.56 ± 2.19, 75.55 ± 4.33, 78.99 ± 4.55, 82.43 ± 0.44, and 93.32 ± 5.76 µM, respectively. Ginsenosides Rb2, Rs1, and Rs2 were found to be inactive at the concentrations tested. To explain the relationship between structure and action, we examined the various capacity of ginsenosides to inhibit HRAR. This inhibitory action enhanced the number of sugar moieties but reduced the ginsenoside molecule. Ginsenosides with three or more sugar molecules, such as Rc, Ra1, Ra2, Rb1, Rb2, Rb3, Rs1, Rs2, Re, and Rd exhibited weak HRAR inhibitory action as compared with the ginsenoside compound K (one sugar residue), Rh1, Rh2, PPT (no sugar moiety), and aglycone PPD. Remarkably, PPT and aglycone PPD exposed solid HRAR inhibition, demonstrating that the absence of a sugar moiety at the C-6, C-3, or C-20 location on the aglycone possibly plays a significant role in ginsenoside action, and the exchange of any sugar residues enhanced the decreasing action. We also detected that the sugar linkage cloud impacted the inhibition of HRAR. Compound K and ginsenosides Rh1 and Rh2 have a sugar linkage at C-20, C-6, and C-3, respectively. It has analogous chemical structures, but HRAR is highly inhibited by compound K, with a glucose linkage at location C-20, then Rh1 or Rh2.

Furthermore, the number and position of ginsenoside hydroxyl groups influenced HRAR inhibition. On the ginsenosides, a hydroxyl group at C-3 or C-20 with a single sugar moiety increased HRAR inhibition. Compound K, ginsenosides Rh2, Rh1, and Rf, (20*S*) ginsenoside Rg3, (20*S*) ginsenoside Rg3, (20*R*) ginsenoside Rg3, (20*S*) ginsenoside Rg2, and (20*R*) ginsenoside Rg2, which contain a single hydroxyl group at the C-3 or C-20 position, were more effective in inhibiting HRAR than those devoid of any hydroxyl group. As a result, it is possible that the presence of a hydroxyl group at the C-3 or C-20 position on the ginsenoside molecule plays a critical role in inhibiting HRAR, whereas the activity decreased when other groups were substituted. Further, we observed the stereoselectivity of the 20(*S*) and 20(*R*) forms of ginsenosides, which are stereoisomers that depend upon the orientation of the C-20 hydroxyl in ginsenosides. As we saw in (20*R*) Rg2, (20*R*) Rg3, (20*S*) Rg2, and (20*S*) Rg3, the stereochemistry difference creates various HRAR inhibitory effects. Both (20*R*) Rg2 and (20*R*) Rg3 inhibited HRAR more effectively than (20*S*) Rg2 and (20*S*) Rg3. These findings suggest that the stereo-structure of the C-20 hydroxyl may alter ginsenosides’ HRAR inhibitory action.

There were significant differences in the inhibitory potencies of ginsenosides against HRAR and RLAR, according to our findings. These results demonstrate clearly that the differences in potency are attributable to the ARIs’ chemical structures, as well as different AR sources, such as the species, organ, and tissue [10]. We found that AR’s susceptibility to various ARIs can exhibit striking differences, depending upon human and animal sources, which may occur, at least in part, because of different degrees of bulk tolerance for the various enzymes [37,38].

### 2.3. Enzyme Kinetics Analysis of RLAR and HRAR Inhibition with Ginsenoside Derivatives

To explain the mode of enzymatic inhibition of ginsenoside derivatives, we performed a kinetics analysis of RLAR and HRAR to determine the representative inhibitors (Table 1). The types of inhibition and the inhibition constants (*K*_ic_) of eight active ginsenosides were investigated using Dixon (Figure 2a–h) and Lineweaver-Burk plots (Figure 2i–p). As shown in Figure 2j–l,p and Table 1, (20*S*) Rg3, (20*R*) Rg3, Rh2, and Rh1 exhibited a competitive mode of RLAR inhibition with an increased *K*_m_ (Michaelis Menten constant) and an unchanged V_max_ (maximal rate) with the increase in concentration, and their respective *K*_ic_ values of 1.97, 3.69, 0.43, and 5.39 µM, respectively. Protopanaxadiol, ginsenoside Rf, and (20*S*) ginsenoside Rg2 demonstrated a mixed type of RLAR inhibition, as when the inhibitor concentration increased, *K*_m_ increased and V_max_ decreased, with *K*_ic_ values of 18.77, 11.17, and 10.77 µM, respectively (Figure 2i,n,o), whereas the *K*_iu_ values were 37.39, 18.60, and 17.04 µM, respectively. For protopanaxatriol, the reduced effect of the V_max_ with unaffected *K*_m_ indicates the non-competitive inhibition mode with a *K_i_* value of 28.88 µM (Figure 2m). Next, kinetics analyses were performed with different concentrations of _DL_-glyceraldehyde substrate and various concentrations of the eight active ginsenosides, as described in the Materials and Methods. To confirm the inhibition pattern, a Lineweaver-Burk plot was drawn using data from the kinetics studies (Figure 3i–p), and the inhibition constants (*K*_ic_) were determined by interpreting the Dixon plots (Figure 3a–h). As shown in Figure 3i,j,m,n, and Table 1, protopanaxadiol, (20*S*) ginsenoside Rg3, compound K, and protopanaxatriol exhibited a competitive mode of HRAR inhibition with an increased *K*_m_ and an unchanged V_max_ with the increase in concentration, and their respective *K*_ic_ values were 0.72, 8.06, 1.59, and 2.61 µM. (20*R*) ginsenoside Rg3, ginsenoside Rh2, and ginsenoside Rh1 demonstrated a mixed type of HRAR inhibition, as the inhibitor concentration increased, *K*_m_ increased and V_max_ decreased, with *K*_ic_ values of 11.08, 3.63, and 3.40 µM, respectively (Figure 3k,l,p), whereas *K*_iu_ had values of 17.99, 9.22, and 8.31 µM, respectively. For the (20*R*) ginsenoside Rg2, the reduced effect of the V_max_ with unaffected *K*_m_ indicates the non-competitive inhibition mode, with a *K*_ic_ value of 11.67 µM (Figure 3o). As the *K_i_* value represents the concentration needed to form an enzyme-inhibitor complex, a lower *K_i_* value may manifest more effective inhibitors against RLAR and HRAR in the development of preventive and therapeutic agents.

### 2.4. Molecular Docking Analysis

Molecular docking studies were performed to predict the binding modes of the compounds investigated. The comparison of docked compounds was carried out with the cognate ligands, which were re-docked in their respective proteins for validation purposes. The crystal (*X*-ray) structures of desired receptors were downloaded from the Protein Data Bank [39,40], and docking studies were performed by LeadIT software (version 2.3.2; Sankt Augustin, Germany) using FlexX features with default settings [41]. The cognate ligands for the human aldose reductases were zenarestat and sulindac with PDB IDs: 1IEI and 3RX2, respectively. To justify the experimental results, binding interactions were explored to gain an in-depth insight and the key interactions of ginsenosides and cognate ligands were demonstrated.

### 2.5. Docking Interactions of Ginsenosides and Cognate Ligand with RLAR

As shown in Figure 4a−j, 3D interactions in the docking simulation took place between ginsenosides and within the RLAR active pocket. The results of the docking analysis of the cognate ligand (sulindac) of RLAR demonstrated that the main amino acids involved in the interactions are Phe122, Cys298, His110, Trp111, Tyr48, Trp20, Trp219, and Val47, which form a substrate binding cavity toward the nicotinamide adenine dinucleotide phosphate (NAP316). These amino acids are responsible for hydrogen bonds, charge–charge interactions, π-alkyl linkages, and π-π T-shaped interactions. The probable binding mode of ginsenosides within the active pocket of aldose reductase revealed similar interactions within the entrance of the hydrophobic cavity and hydrophilic head, thus stabilizing the ginsenosides. The key interactions and amino acid residues are reported briefly in Table 2. The ginsenosides exhibited conventional hydrogen bonds with Tyr48, Phe121, Phe122, Ser302, Glu120, Val47, Trp20, and Tyr48, in addition to the amino acids reported earlier. Moreover, His110 and Tyr209 demonstrated π-π stacked interactions. The other contacts ginsenosides demonstrated were π-sulfur with Cys298, π-lone pair by Trp219, π-sigma by Ile260, and π-alkyl interaction with Trp20, Trp79, Trp111, Trp219, Phe122, His46, and Phe121. Moreover, the binding modes of ginsenosides manifested a pose similar to that of sulindac, and noticeable interactions with the proteins were monitored. The detailed analysis of the binding modes revealed that the network of H-bonds with the di- and trihydroxy groups, hydrophobic interactions with the alkyl chain, and π-π interactions with the hexadecahydrocyclopenta[α]phenanthrene nucleus of the ginsenosides extended to the entrance cavity of the hydrophobic domain and thus stabilized the binding. Most of the ginsenosides were inclined toward the hydrophilic head and exhibited the most conspicuous interaction with His110 and Trp111. The R1 and R2 substituents occupied the hydrophobic cavity and were oriented toward the catalytic cavity of 3RX2, which produced the best-oriented conformation. These interactions can be instrumental factors in the binding of ginsenosides with protein in the presence of cofactor NAP316. In addition to strong hydrogen bonds, some of the compounds exhibited hydrophobic interactions, as illustrated in the 2D interaction diagrams (Appendix A). Hence, our results revealed that the putative binding modes of ginsenosides are extended toward the entrance pocket and within the catalytic domain of the receptors.

### 2.6. Docking Interactions of Ginsenosides and Cognate Ligand with HRAR

The probable binding mode of the cognate ligand zenarestat demonstrated several crucial interactions, including H-bonds with Trp111, Cys298, Tyr309, and Tyr48 within the active site of HRAR, while various key amino acids, such as Tyr48, Trp20, Trp219, and Trp111 exhibited π-π T-shaped and π-π stacked interactions. Moreover, alkyl and π-alkyl linkages were noted with Tyr309, Trp20, Trp111, Phe122, Tyr48, Leu300, Gln49, Tyr48, and Val47. In addition, zenarestat exhibited a π-sigma interaction with Trp20, fluorine interaction with His110, and π-sulfur interactions with Cys298. The 7-chloro-2,4-dioxoquinazolin-1-yl moiety of zenarestat was aligned with the cofactor, NAP350; however, the 4-bromo-2-fluorophenyl part of zenarestat was inclined toward the distinct sites of the receptor’s hydrophilic pocket. The key interactions and amino acid residues are reported briefly in Table 3. The interactions revealed that most of the ginsenosides demonstrated a significant number of hydrophobic interactions with several key amino acid residues in addition to vital hydrogen bonds. As these structures are relatively complex compared to zenarestat, part of the ginsenosides was found to exhibit interactions with the catalytic site of the enzyme and exhibited multiple interactions, including alkyl, π-alkyl, and π-π stacked contacts, and all of the compounds were inclined toward the cofactor nicotinamide adenine dinucleotide phosphate (NAP350). Other key interactions of docked ginsenosides within the binding site of the aldose reductase included π-donor with Tyr48, π-π T-shaped interactions with Trp20 near NAP350, attractive charges with amino acid Lys77, and π-sulfur with Cys298. Moreover, H-bonds with Gln49, Val47, Asp216, Ile260, Gln183, and Lys77 were noticed near the hydrophobic pocket of the enzyme. The most promising interactions the docked ginsenosides demonstrated within the active pocket of aldose reductase (HRAR) are shown in Figure 5a–j.

The 3D interaction in Figure 5a–j revealed the binding of ginsenosides with the hydrophilic cavity with multiple H-bonding, alkyl linkages, π-alkyl interactions, and π-π stacking near the NAP350. As shown in the figures, the ginsenosides adopted the conformations at the entrance site, active pocket, and catalytic cleft of 1IEI well by exhibiting multiple bonds of several types attributable to many hydroxyl groups (H-bonding), the main hexadecahydrocyclopenta[α]phenanthrene nucleus (π-π T-shaped, π-alkyl and π-π stacked interactions), and the alkene chain (alkyl linkages). Therefore, the binding of ginsenosides within the aldose reductase (1IEI) was stabilized by the presence of several types of interactions and demonstrated 3D-binding interactions similar to those zenarestat exhibited. In addition to strong hydrogen bonds, some of the ginsenosides exhibited hydrophobic interactions, as shown in the 2D interaction diagrams (Appendix A).

### 2.7. Inhibitory Activities of Active Ginsenosides on Sorbitol Accumulation

We studied the effects of the 10 active ginsenosides on sorbitol accumulation in rat lenses to demonstrate their efficacy in ex vivo. Sorbitol can be created faster than fructose can be processed in a hyperglycemic state, resulting in sorbitol buildup. The polarity of the sugar alcohol promotes intracellular accumulation because it prevents easy membrane penetration and subsequent removal via diffusion. Thus, the intracellular accumulation of a polar sugar alcohol can cause a hyperosmotic effect, resulting in a fluid infusion to counteract the osmotic gradient. Fluid influx has been demonstrated to cause alterations in membrane permeability and the start of cellular disease [42,43]. Rat lenses treated in 25 mM glucose for 6 days exhibited enhanced intracellular sorbitol accumulation. As shown in Table 4, the ginsenosides protopanaxadiol, (20*S*) ginsenoside Rg3, (20*R*) ginsenoside Rg3, ginsenoside Rh2, compound K, protopanaxatriol, ginsenoside Rf, (20*S*) ginsenoside Rg2, (20*R*) ginsenoside Rg2, and ginsenoside Rh1 inhibited sorbitol accumulation effectively by 96.51%, 78.48%, 48.25%, 97.09%, 73.25%, 94.76%, 83.72%, 57.55%, 51.74%, and 94.18%, respectively, at a concentration of 5 μg/mL. The positive control (quercetin) inhibited sorbitol accumulation in rat lenses by 87.20% and reduced sorbitol levels in a culture medium containing a high glucose concentration. Aldose reductase inhibitors have already been demonstrated to prevent sorbitol accumulation and reduce diabetic consequences [44]; therefore, edible ginseng extracts and their major active ingredients (ginsenosides) are a promising therapeutic aid in the treatment of diabetes complications, as ginsenosides may exert comprehensive inhibitory effects against diabetic complications via the AR-polyol pathway and the insulin resistant and AGE formation systems, which are implicated highly in oxidative stress. In our previous study, ginsenoside derivatives exhibited AGE formation inhibitory activity [33], and antioxidant activities in our antioxidant assays, including the reactive oxygen species (ROS) and ONOO^−^ systems, and improved glucose uptake in insulin-resistant HepG2 cells [34]. Thus, our previous and current results suggest that ginsenosides may possess good inhibitory activities against diabetic complications, as suggested by their significant antioxidant activity.

## 3. Materials and Methods

### 3.1. Chemicals and Reagents

_DL_-glyceraldehyde dimer, adenine dinucleotide phosphate (NADPH), β-nicotinamide bovine serum albumin (BSA), and quercetin were purchased from Sigma-Aldrich (St. Louis, MO, USA). Human recombinant AR (0.4 units) was purchased from Wako Chemicals (Osaka, Japan). Sodium azide was purchased from Junsei Chemical Co. (Tokyo, Japan). Ginsenoside Rb2 (>95%), ginsenoside Rb3 (>95%), protopanaxadiol (purity >85%), ginsenoside Rf (>95%), ginsenoside Rc (>98%), protopanaxatriol (>96%), ginsenoside Re (>97%), (20*S*) ginsenoside Rg3 (>98%), ginsenoside Rd (>95%), ginsenoside Rg1(>90%), ginsenoside Rh2 (>97%), compound K (>96%), sodium hydroxide, ginsenoside Rb1 (>98%), quercetin (>95%), ginsenoside Rh1(>90%), glucose, dimethyl sulfoxide (DMSO), and sorbitol were purchased from Sigma-Aldrich Co. (St Louis, MO, USA). (20*R*) ginsenoside Rg3 (purity > 96%), (20*S*) ginsenoside Rg2 (>90%), ginsenoside Ra1 (>96%), ginsenoside Ra2 (>98%), ginsenoside Rs1 (>97%), ginsenoside Rs2 (>90%), (20*R*) ginsenoside Rg2 (>99%) were purchased from Selleckchem Co. (Cedarlane, ON, Canada). Unless otherwise noted, all additional chemicals and solvents were reagent grade and purchased from Merck (Darmstadt, Germania), Fluka (Buchs, Switzerland), Duksan Pure Chemical Co. (Ansan, South Korea), or Sigma-Aldrich Co.

### 3.2. Assay for RLAR Inhibitory Activity

A rat lens homogenate was prepared with minor modifications according to the method reported by Hayman et al. [45]. Briefly, the lenses were removed from the eyes of Sprague-Dawley rats weighing 250–280 g. They were homogenized in sodium phosphate buffer (pH 6.2) that was prepared with dibasic sodium phosphate (Na_2_HPO_4_∙H_2_O, 0.66 g) and monobasic sodium phosphate (NaH_2_PO_4_∙H_2_O, 1.27 g) in 100 mL of double-distilled H_2_O. The homogenate was clarified by centrifugation at 10,000 rpm at 4 °C for 20 min and the resultant supernatant was frozen until use. A crude AR homogenate with a specific activity of 6.5 U/mg was used in all enzyme inhibition evaluations. The reaction solution consisted of 620 µL of 100 mM sodium phosphate buffer (pH 6.2), 90 µL of AR homogenate, 90 µL of 1.6 mM NADPH, and 9 µL of sample, while the substrate included 90 µL of 50 mM _DL_-glyceraldehyde. Quercetin, a well-known AR inhibitor, was used as a control. AR activity was determined by measuring the decrease in NADPH absorption at 340 nm over a 4 min period on an Ultrospec^®^2100pro UV/Visible spectrophotometer. SWIFT II Applications software (Amersham Biosciences, NJ, USA) was used for all data analyses. The inhibition percentage (%) was calculated as follows in Equation (1):[1 − (∆Asample/min − ∆Ablank/min)/(∆Acontrol/min − ∆Ablank/min)] × 100(1)
in which ∆Asample/min represents the test sample and substrate’s reduction in absorbance over 4 min, and ∆A control/min represents the same reduction in absorbance, but with 100% DMSO (dimethyl sulfoxide) rather than the test sample. The 50% inhibition concentration is expressed as the mean ± SEM.

### 3.3. HRAR Inhibition Assay

HRAR inhibition was studied according to Nishimura et al. [38]. AR action was studied by measuring the reduction in NADPH absorption at 340 nm over a period of 1 min on a UV/visible spectrophotometer (Ultrospec^®^2100pro, Biochrom, Holliston, MA, USA). For HRAR inhibition, the HRAR enzyme (0.4 units), was used in all enzyme inhibition evaluations. ARIs (aldose reductase inhibitors) were applied as positive controls (quercetin and zenaresta). The inhibition percentage (%) was measured as designated in the RLAR assay. The DA sample/min value characterizes the decrease in absorbance for 1 min with the test samples and substrate. The ability of each sample to inhibit HRAR was observed as an IC_50_ value (µM) and measured from the log-dose inhibition curve.

### 3.4. Determination of Kinetics Parameters of RLAR and HRAR Inhibition via Lineweaver-Burk and Dixon Plots

Lineweaver-Burk and Dixon plots [1/enzyme velocity (1/*V*) versus 1/substrate concentration (1/[S])] was produced to determine the kinetic mechanisms of inhibition [46,47,48], which demonstrated the complementary results. Enzyme inhibition with different concentrations (100, 50, and 25 mM) was assessed by monitoring the effects of different concentrations of the substrate. The following concentrations of test ginsenosides were used in the RLAR kinetics analysis: 0, 2, 10, and 50 µM for protopanaxadiol; 0, 0.5, 2.5, and 10 µM for (20*S*) ginsenoside Rg3; 0, 0.5, 2.5, and 10 µM for (20*R*) ginsenoside Rg3; 0, 0.1, 0.5, and 2.5 µM for ginsenoside Rh2; 0, 2, 10, and 50 µM for protopanaxatriol; 0, 0.8, 4, and 20 µM for ginsenoside Rf; 0, 0.8, 4, and 20 µM for (20*S*) ginsenoside Rg2, and 0, 0.5, 2.5, and 10 µM for ginsenoside Rh1. Using Lineweaver-Burk double reciprocal plots, the inhibition type was determined using various concentrations of _DL_-glyceraldehyde (5, 10, and 20 mM) for HRAR as a substrate in the presence of different concentrations of the active test ginsenosides. The concentrations of test ginsenosides in the HRAR kinetics analysis were as follows: 0, 0.1, 0.5, and 2.5 µM for protopanaxadiol; 0, 0.5, 2, and 10 µM for (20*S*) ginsenoside Rg3; 0, 1, 5, and 20 µM for (20*R*) ginsenoside Rg3; 0, 0.5, 2.5, and 10 µM for ginsenoside Rh2; 0, 0.1, 0.5, and 2.5 µM for compound K; 0, 1, 5, and 10 µM for protopanaxatriol; 0, 0.5, 4, and 20 µM for (20*R*) ginsenoside Rg2, and 0, 0.2, 2, and 5 µM for ginsenoside Rh1. The Dixon plot is a graphical method [1/enzyme velocity (1/*V*)] against inhibitor concentration [I] used to determine the type of enzyme inhibition; it was used to identify the dissociation or inhibition constant (*K_i_*) for the enzyme-inhibitor complex. Dixon plots (single reciprocal plots) for RLAR and HRAR inhibition were obtained in the presence of 100, 50, and 10 mM RLAR, and 20, 10, and 5 mM HRAR _DL_-glyceraldehyde substrate. Thus, the type of enzyme inhibition was determined by interpreting the Lineweaver-Burk plots, and the inhibition constants (*K_i_*) were determined by interpreting the Dixon plots. 

### 3.5. Lens Culture and Intracellular Sorbitol Measurement

Lenses were isolated from Sprague-Dawley rats through cultures with sterile conditions and an atmosphere (95% air and 5% CO_2_ at 37 °C for 6 d) in TC-199 medium comprising 100 U/mL penicillin, 15% fetal bovine serum, and 0.1 mg/mL streptomycin. The ginsenosides were liquified in DMSO. The lenses were separated into three groups with a medium containing RLAR-active ginsenosides and 25 mM glucose. Each lens was placed in a medium (2.0 mL). The sorbitol was calculated by HPLC [49].

### 3.6. Molecular Docking Simulation in RLAR and HRAR Inhibition

To perform the molecular docking analysis, structures of the target proteins HRAR (PDB: 1IEI) [39], and RLAR (PDB: 3RX2) [40] were obtained from the Protein Data Bank. The binding pocket and binding poses of docked compounds were investigated together with cognate ligands, as reported previously [7,33]. The active pocket of each receptor was identified using the SiteFinder feature within the Molecular Operating Environment (version 2019. 0201, Montreal, QC, Canada) [50]. The structures of the target proteins/receptors were protonated by the AMBER99 force field followed by the energy minimization at RMSD gradient of 0.05 kcal/mol [40]. Three-dimensional (3D) structures of the ginsenosides were prepared using the builder utility of MOE builder default parameters, followed by energy minimization; subsequently, the interactions of ginsenosides and the proteins’ cognate ligands were examined within the active pocket, and the results were exhibited in the form of 3D poses [50]. Energy minimizations for these structures were performed using the MMFF94x force field at the RMSD of 0.01 kcal/mol Å in MOE [51], and cognate ligands were downloaded from the RCSB Protein Data Bank. The molecular docking analysis was performed using LeadIT software (version 2.3.2, Sankt Augustin, Germany) by setting default parameters [41]. Before the analysis, a validation of the methodology was performed by docking the cognate ligands within their respective proteins followed by docking the ginsenosides within the active site of the target protein. The top 50 binding poses were selected and inspected by the HYDE visual affinity. The HYDE assessment is helpful in selecting the final poses based upon favorable interactions for each ligand [52]. The poses with the lowest binding score and the most favorable affinity were selected and visualized with the Discovery Studio Visualizer (version 21.1.0.20298; San Diego, CA, USA) [53].

### 3.7. Statistical Analysis

All experimental results were expressed as the mean ± SEM of triplicate experiments. Statistically significant values were analyzed using analysis of variance (ANOVA) and Duncan’s test (Systat Inc., Evanston, IL, USA). A *p*-value < 0.05 was considered statistically significant.

## 4. Conclusions

In summary, this study investigated the RLAR and HRAR inhibitory activities of ginsenoside derivatives. Ginsenosides demonstrated high inhibitory activities in two in vitro assays, and their potentially beneficial effects on diabetic complications make ginsenosides an excellent ingredient for functional foods. Additionally, the eight most active ginsenosides demonstrated potent inhibitory activities on sorbitol accumulation in isolated rat lenses. Our data indicated as well that aglycone PPD or PPT, and a hydroxyl group with a single sugar moiety containing ginsenosides, inhibit aldose reductase activity in vitro, and the mode of action depends upon the ginsenosides’ structure. These structure−function relations could be useful in designing new aldose reductase inhibitors based upon ginsenosides. Collectively, this study together with our previous finding on the AGE formation inhibitory activity, suggests that ginsenoside derivatives can be potent functional food ingredients as RLAR and AGE inhibitors and can be used as a naturotherapy for diabetic complications. Further investigation to clarify their beneficial/harmful effects in vivo are underway in our laboratory.

## Figures and Tables

**Figure 1 molecules-27-02134-f001:**
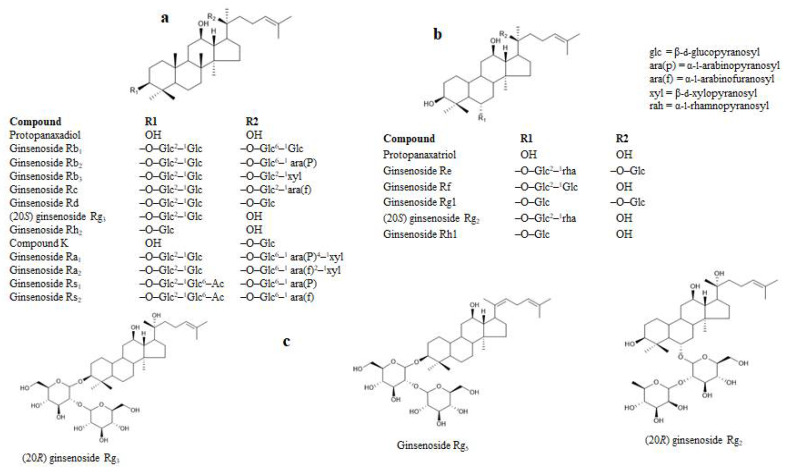
Structure of selected ginsenosides. (**a**) PPDs. (**b**) PPTs. (**c**) Derivatives of PPDs and PPTs. Glc, β-d-glucopyranosyl; ara(P), α-l-arabinopyranosyl; ara(f), α-l-arabinofuranosyl; Xyl, β-dxylopyranosyl; rah, α-l-rhamnopyranosyl.

**Figure 2 molecules-27-02134-f002:**
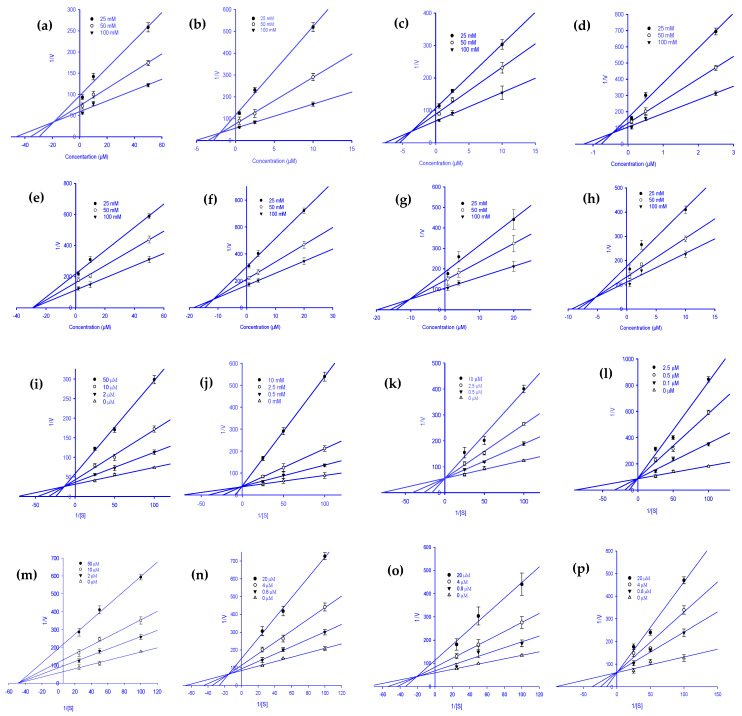
Dixon plots for RLAR inhibition by various ginsenoside derivatives. Protopanaxadiol (**a**), (20S) ginsenoside Rg3 (**b**), (20*R*) ginsenoside Rg3 (**c**), ginsenoside Rh2 (**d**), protopanaxatriol (**e**), ginsenoside Rf (**f**), (20*S*) ginsenoside Rg2 (**g**), and ginsenoside Rh1 (**h**) were tested in the presence of 25 mM (●), 50 mM (○), and 100 mM (▼) substrate (_DL_-glyceraldehyde). Lineweaver-Burk plots for RLAR inhibition were analyzed in the presence of the following ginsenoside concentrations: 0 µM (Δ), 2 µM (▼), 10 µM (○), and 50 µM (●) for protopanaxadiol (**i**); 0 µM (Δ), 0.5 µM (▼), 2.5 µM (○), and 10 µM (●) for (20*S*) ginsenoside Rg3 (**j**); 0 µM (Δ), 0.5 µM (▼), 2.5 µM (○), and 10 µM (●) for (20*R*) ginsenoside Rg3 (**k**); 0 µM (Δ), 0.1 µM (▼), 0.5 µM (○), and 2.5 µM (●) for ginsenoside Rh2 (**l**); 0 µM (Δ), 2 µM (▼), 10 µM (○), and 50 µM (●) for protopanaxatriol (**m**); 0 µM (Δ), 0.8 µM (▼), 4 µM (○), and 20 µM (●) for ginsenoside Rf (**n**); 0 µM (Δ), 0.8 µM (▼), 4 µM (○), and 20 µM (●) for (20*S*) ginsenoside Rg2 (**o**); 0 µM (Δ), 0.5 µM (▼), 2.5 µM (○); and 10 µM (●) for ginsenoside Rh1 (**p**).

**Figure 3 molecules-27-02134-f003:**
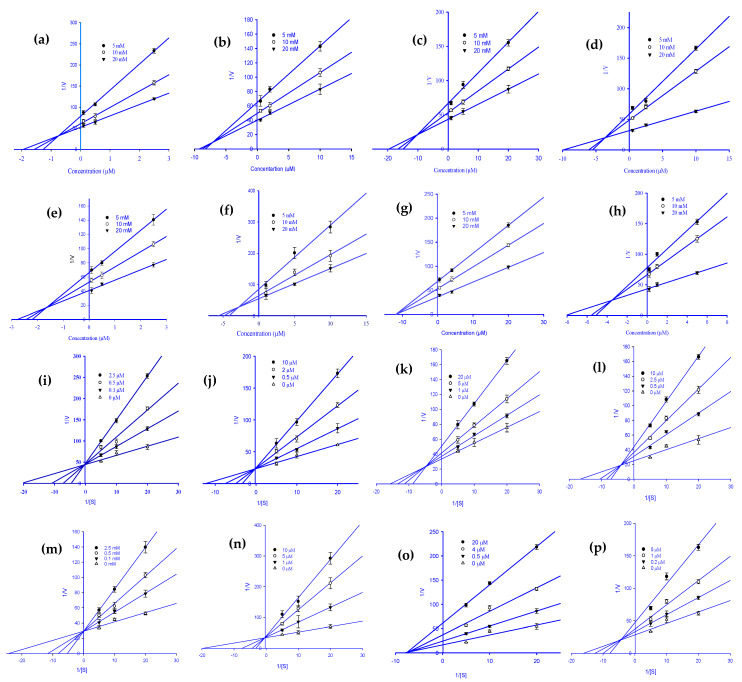
Dixon plots for HRAR inhibition by various ginsenoside derivatives. Protopanaxadiol (**a**), (20*S*) ginsenoside Rg3 (**b**), (20*R*) ginsenoside Rg3 (**c**), ginsenoside Rh2 (**d**), compound K (**e**), protopanaxatriol (**f**), (20*R*) ginsenoside Rg2 (**g**), and ginsenoside Rh1 (**h**) were tested in the presence of 5 mM (●), 10 mM (○), and 20 mM (▼) substrate (_DL_-glyceraldehyde). Lineweaver–Burk plots for HRAR inhibition were analyzed in the presence of the following ginsenoside concentrations: 0 µM (Δ), 0.1 µM (▼), 0.5 µM (○), and 2.5 µM (●) for protopanaxadiol (**i**); 0 µM (Δ), 0.5 µM (▼), 2 µM (○), and 10 µM (●) for (20*S*) ginsenoside Rg3 (**j**); 0 µM (Δ), 1 µM (▼), 5 µM (○), and 20 µM (●) for (20*R*) ginsenoside Rg3 (**k**); 0 µM (Δ), 0.5 µM (▼), 2.5 µM (○), and 10 µM (●) for ginsenoside Rh2 (**l**); 0 µM (Δ), 0.1 µM (▼), 0.5 µM (○), and 2.5 µM (●) for compound K (**m**); 0 µM (Δ), 1 µM (▼), 5 µM (○), and 10 µM (●) for protopanaxatriol (**n**); 0 µM (Δ), 0.5 µM (▼), 4 µM (○), and 20 µM (●) for (20R) ginsenoside Rg2 (**o**); 0 µM (Δ), 0.2 µM (▼), 1 µM (○); and 5 µM (●) for ginsenoside Rh1 (**p**).

**Figure 4 molecules-27-02134-f004:**
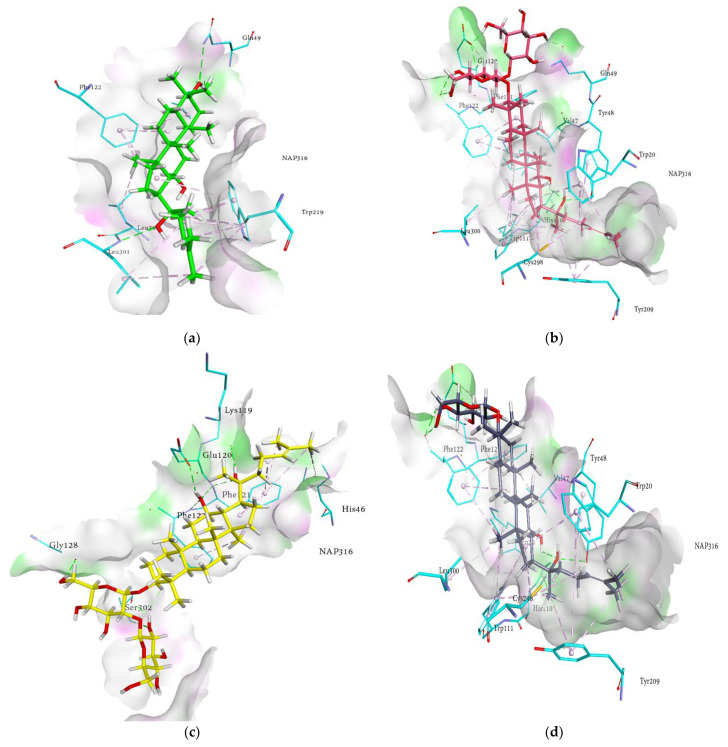
Molecular docking models for RLAR inhibition by ginsenosides and diagrams of ligand 3D interactions and the major binding sites of the eight ginsenosides in the RLAR active site; protopanaxadiol (**a**), (20*S*) ginsenoside Rg3 (**b**), (20*R*) ginsenoside Rg3 (**c**), ginsenoside Rh2 (**d**), protopanaxatriol (**e**), ginsenoside Rf (**f**), (20*S*) ginsenoside Rg2 (**g**), ginsenoside Rh1 (**h**), quercetin (**i**), and sulindac (**j**). The interactions are shown by green (conventional hydrogen bonding), yellow (π-sulfur interactions), tea pink (π-π T-shaped and π-π stacked interactions), cyan (fluorine) and yellow (π-sulfur).

**Figure 5 molecules-27-02134-f005:**
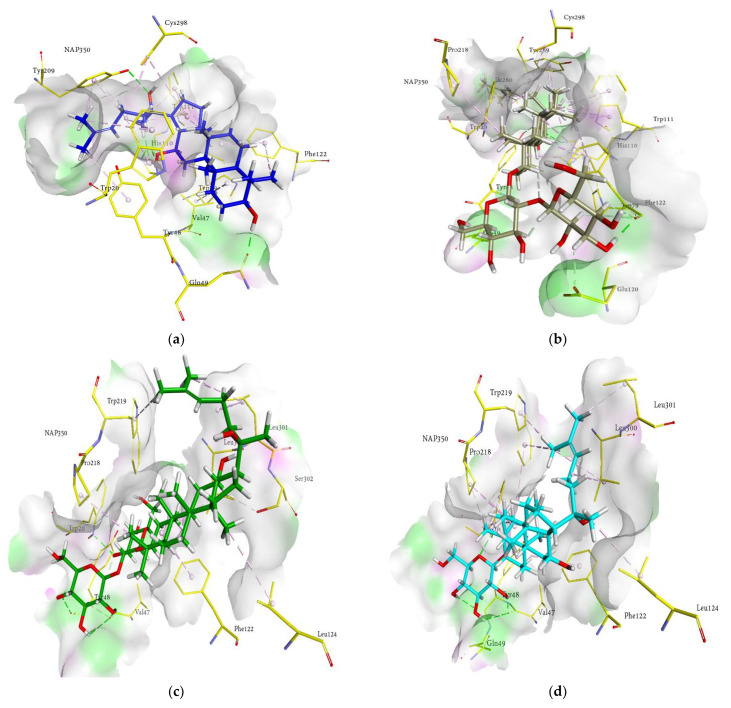
Molecular docking models for HRAR inhibition by ginsenosides and diagrams of ligand 3D interactions and the major binding sites of the eight ginsenosides in the HRAR active site; protopanaxadiol (**a**), (20*S*) ginsenoside Rg3 (**b**), (20*R*) ginsenoside Rg3 (**c**), ginsenoside Rh2 (**d**), compound K (**e**), protopanaxatriol (**f**), (20*R*) ginsenoside Rg2 (**g**), ginsenoside Rh1 (**h**), quercetin (**i**), and zenaresta (**j**). The interactions are shown by green (conventional hydrogen bonding), yellow (π-sulfur interactions), tea pink (π-π T shaped and π-π stacked interactions), cyan (fluorine), and yellow (π-sulfur).

**Table 1 molecules-27-02134-t001:** The in vitro inhibitory activities of ginsenosides on RLAR and HRAR.

Compounds	RLAR			HRAR		
	IC_50_ (µM) ^a^	*K_i_* (µM) ^b^	Mode of Inhibition ^c^	IC_50_ (µM) ^a^	*K_i_* (µM) ^b^	Mode of Inhibition ^c^
Protopanaxadiol	21.38 ± 2.45	18.77 (*K*_ic_), 37.39 (*K*_iu_)	Mixed type	0.36 ± 0.1	0.72	Competitive type
Ginsenoside Rb1	103.11 ± 7.45			93.32 ± 5.76		
Ginsenoside Rb2	121.12 ± 3.43			>100		
Ginsenoside Rb3	117.43 ± 6.89			78.99 ± 4.55		
Ginsenoside Rc	>200			56.56 ± 2.19		
Ginsenoside Rd	89.58 ± 2.99			37.45 ± 1.33		
(20*S*) ginsenoside Rg3	1.25 ± 0.28	1.97	Competitive type	9.92 ± 0.56	8.06	Competitive type
(20*R*) ginsenoside Rg3	4.28 ± 0.31	3.69	Competitive type	8.67 ± 0.87	11.08 (*K*_ic_), 17.99 (*K*_iu_)	Mixed type
Ginsenoside Rg5	54.47 ± 1.22			38.56 ± 2.91		
Ginsenoside Rh2	0.67 ± 0.01	0.43	Competitive type	7.44 ± 0.55	3.63 (*K*_ic_), 9.22 (*K*_iu_)	Mixed type
Compound K	41.48 ± 3.99			2.23 ± 0.54	1.59	Competitive type
Ginsenoside Ra1	>200			75.55 ± 4.33		
Ginsenoside Ra2	178.39 ± 7.39			82.43 ± 0.44		
Ginsenoside Rs1	142.77 ± 3.77			>100		
Ginsenoside Rs2	149.34 ± 4.22			>100		
Protopanaxatriol	27.88 ± 1.19	28.88	Non-competitive type	1.43 ± 0.14	2.61	Competitive type
Ginsenoside Re	81.27 ± 2.18			43.45 ± 3.11		
Ginsenoside Rf	11.29 ± 1.49	11.17(*K*_ic_), 18.60 (*K*_iu_)	Mixed type	19.45 ± 1.55		
Ginsenoside Rg1	49.48 ± 1.88			27.56 ± 2.12		
(20*S*) ginsenoside Rg2	14.38 ± 0.99	10.77 (*K*_ic_), 17.04 (*K*_iu_)	Mixed type	15.67 ± 1.05		
(20*R*) ginsenoside Rg2	29.38 ± 2.33			13.66 ± 0.99	11.67	Non-competitive
Ginsenoside Rh1	7.28 ± 0.27	5.39	Competitive type	4.66 ± 0.34	3.40 (*K*_ic_), 8.31 (*K*_iu_)	Mixed type
Quercetin ^d^	4.88 ± 0.71	-	-	3.11 ± 0.22	-	
Zenaresta ^e^	-	-	-	0.69 ± 0.11	-	

^a^ The 50% inhibition concentration (µM) is calculated from a log-dose inhibition curve and expressed as the mean ± S.E.M. of triplicate experiments. ^b^ *K*_ic_ (binding constants of inhibitor with free enzyme) and *K*_iu_ (binding constants of inhibitor with enzyme–substrate complex) values were determined by Dixon plots. ^c^ Inhibition type was determined by interpretation of the Lineweaver-Burk plot. ^d,e^ Positive controls were used in the respective assays.

**Table 2 molecules-27-02134-t002:** Binding energies and binding interactions of ginsenosides with RLAR using the LeadIT docking program (FlexX) and visualization by Discovery Studio Visualizer.

Compounds	Docked Energy (kcal/mol)	Hydrogen Bond Interactions (No. of H-Bond)	Hydrophobic Interactions
(20*R*) ginsenoside Rg3	−6.10	Lys119 (2.15 Å), Glu120 (2.29 Å), Phe121 (2.44 and 2.45 Å), Phe122 (2.66 Å), Gly128 (2.08 Å), Ser302 (3.14 and 1.91 Å)	His46 (π-alkyl 5.05 Å), Phe121 (π-alkyl 5.21, 4.38, 5.26 and 5.34 Å), Phe122 (π-alkyl 5.19 Å)
(20*S*) ginsenoside Rg2	−10.31	Val47 (2.09 Å), Glu120 (1.94 Å), Phe121 (2.84 Å), Phe122 (3.04 and 2.11 Å)	Phe122 (π-alkyl 4.84, 4.47, 3.10 and 5.20 Å), Leu301 (Alkyl 5.14 Å)
(20*S*) ginsenoside Rg3	−14.56	Tyr48 (2.88 Å), Gln49 (1.65 Å), Glu120 (2.55 and 1.54 Å), Phe122 (2.15 and 2.09 Å)	Trp20 (π-alkyl 4.73, 4.99, 4.08, 4.06 and 5.21 Å), Val47 (Alkyl 5.38 and 4.86 Å), Trp79 (π-alkyl 5.40 Å), Trp111 (π-alkyl 4.50, 5.41, 5.36 and 4.84 Å), Phe122 (π-alkyl 5.32, 4.91, 4.76 and 4.37 Å), Tyr209 (π-alkyl 4.88 Å), Trp219 (π-alkyl 5.45 Å), Cys298 (Alkyl 4.40 Å), Leu300 (Alkyl 4.65 Å)
Ginsenoside Rf	−9.44	Val47 (1.82 Å), Gln49 (2.64 Å), Glu120 (2.15 Å), Phe121 (2.85 Å), Phe122 (2.20 and 2.89 Å), Ser302 (2.48 Å)	Phe122 (π-alkyl 3.02, 4.60 and 5.20 Å), Trp219 (π-alkyl 4.00, 5.07 and 5.47 Å, π-lone pair 2.65 Å), Leu301 (Alkyl 4.72 Å)
Ginsenoside Rh1	−17.64	Val47 (1.78 Å), Gln49 (2.82 Å), Glu120 (2.19 Å), Phe121 (2.82 Å), Phe122 (2.05 and 2.87 Å)	Val47 (Alkyl 5.49 Å), Phe122 (π-alkyl 2.86, 4.19, 4.71 and 4.97 Å), Trp219 (π-alkyl 3.99, 4.27, 5.30 and 5.33 Å), Ala299 (Alkyl 3.48 Å), Leu301(Alkyl 4.46 Å)
Ginsenoside Rh2	−17.16	Tyr48 (2.88 Å), Glu120 (2.52 Å), Phe122 (2.09 and 2.18 Å)	Trp20 (π-alkyl 4.06, 4.08, 4.71, 5.21 Å), Val47 (Alkyl 4.86 and 5.38 Å), Trp79 (π-alkyl 5.40 Å), Trp111 (π-alkyl 4.50, 4.84, 5.36 and 5.41 Å), Phe122 (π-alkyl 5.32, 4.91, 4.37, 4.76 Å), Tyr209 (π-alkyl 4.86 Å), Trp219 (π-alkyl 5.45 Å), Cys298 (Alkyl 4.40 Å), Leu300 (Alkyl 4.65 Å)
Protopanaxadiol	−13.74	Val47 (1.83 Å), Gln49 (3.28 Å), Leu301 (2.75 Å)	Trp20 (π-alkyl 5.49 Å), Phe122 (π-alkyl 3.30, 4.04, 4.67, 5.34 Å), Trp219 (π-alkyl 4.94, 4.98 and 5.00 Å), Leu300 (Alkyl 5.22 Å), Leu301 (Alkyl 4.67 Å)
Protopanaxatriol	−9.86	Val47 (1.67 Å), Gln49 (3.16 Å), Leu301 (2.93 Å)	Trp20 (π-alkyl 5.40 Å), Phe122 (π-alkyl 3.24 and 5.31 Å), Trp219 (π-alkyl 4.08, 4.53, 5.34 and 4.95 Å), Ala299 (Alkyl 3.93 Å), Leu301 (Alkyl 3.90 Å)
Quercetin	−31.20	Thr19 (2.81 Å), Asp43 (2.32 Å), Tyr48 (2.15 Å), Trp111 (2.65 Å), Ser210 (2.27, 2.62 and 2.74 Å), Ile260 (2.09 Å)	His110 (π-π T shaped 4.87 Å), Tyr209 (π-π T shaped 5.38 Å)Ile260 (π-sigma 3.98 Å), Cys298 (π-sulfur 4.36 and 5.78 Å)

**Table 3 molecules-27-02134-t003:** Binding energies and binding interactions of ginsenosides with HRAR using the LeadIT docking program (FlexX) and visualization by Discovery Studio Visualizer.

Compounds	Docked Energy (kcal/mol)	Hydrogen Bond Interactions (No. of H-Bond)	Hydrophobic Interactions
(20*R*) ginsenoside Rg2	−4.09	Gln49 (2.11 Å), Pro218 (1.77 Å), Ser302 (3.17 Å)	Phe122 (π-alkyl 3.96, 4.11 and 5.17 Å), Trp219 (π-alkyl 5.15, 5.01, 3.36 and 5.38 Å), Ala299 (Alkyl 4.23 Å), Leu300 (Alkyl 4.60 Å)
(20*R*) ginsenoside Rg3	−5.96	Trp20 (2.44 Å), Val47 (3.02 Å) Tyr48 (2.15 Å)	Phe122 (π-alkyl 4.54 Å), Leu124 (Alkyl 5.11 Å), Pro218 (Alkyl 4.60 Å), Trp219 (π-alkyl 5.15 and 4.69 Å), Leu300 (Alkyl 5.38 Å), Leu301 (Alkyl 5.27 Å)
(20*S*) ginsenoside Rg3	−6.88	Tyr48 (2.58 Å), Gln49 (1.72 Å), His110 (2.41 Å), Phe122 (1.67 Å)	Trp20 (π-alkyl 5.41, 4.98, 3.70, 4.53, 5.07 and 4.78 Å), Trp79 (π-alkyl 5.08 Å), Trp111 (π-alkyl 4.37, 5.11, 4.19 and 4.66 Å), Phe122 (π-alkyl 5.22, 4.60, 4.33 and 4.17 Å), Tyr209 (π-alkyl 4.93 Å), Pro218 (Alkyl 4.81Å), Trp219 (π-alkyl 4.82 Å), Ile260 (Alkyl 5.29 Å), Cys298 (Alkyl 4.61Å)
Compound K	−7.83	Val47 (1.90 Å), Tyr48 (1.92 Å) Gln49 (3.31 and 3.14 Å), Trp111 (2.88 Å)	Trp20 (π-alkyl 4.50, 4.86, 4.66, 4.76, 4.55, 4.61 and 4.33 Å), Val47 (Alkyl 4.82 and 4.10 Å), Trp79 (π-alkyl 4.55 Å), Trp111 (π-alkyl 5.44, 5.00 and 4.69 Å), Phe121 (π-alkyl 4.18 Å), Phe122 (π-alkyl 5.38, 4.48 and 4.29 Å), Pro218 (Alkyl 5.46, 4.00 and 5.05 Å), Trp219 (π-alkyl 5.17 Å), Cys298 (Alkyl 4.06 Å)
Ginsenoside Rh1	−7.47	Val47 (2.70 Å), Tyr48 (2.27 Å) Gln49 (3.07 Å), Cys298 (2.27 and 1.67 Å)	Trp20 (π-alkyl 5.18 Å), Pro23 (Alkyl 4.34 and 4.73 Å), Pro24 (Alkyl 3.92 Å), Val47 (Alkyl 4.15 and 4.46 Å), Phe122 (π-alkyl 4.68, 4.50 and 4.72 Å), Pro218 (Alkyl 5.41, 5.00 and 4.39 Å), Trp219 (π-alkyl 5.23 Å) Leu300 (Alkyl 4.12 Å)
Ginsenoside Rh2	−9.63	Trp20 (2.66 Å), Val47 (4.76 and 3.05 Å), Tyr48 (2.10 Å)	Trp20 (π-alkyl 4.35, 4.69, 4.95 and 4.84 Å), Phe122 (π-alkyl 3.77 and 5.31Å), Pro218 (Alkyl 4.56, 3.45 and 5.28 Å), Trp219 (π-alkyl 4.63, 5.45 and 4.27 Å), Leu300 (Alkyl 4.60 Å), Leu301 (Alkyl 4.39 Å)
Protopanaxadiol	−8.62	Gln49 (1.89 Å), Tyr209 (2.97 Å)	Trp20 (π-alkyl 5.17, 4.00, 4.66, 3.94 and 4.60 Å), Val47 (Alkyl 4.36 Å), Tyr48 (π-alkyl 4.94 Å), Trp79 (π-alkyl 4.23 Å), Trp111 (π-alkyl 5.07, 4.53, 5.10 and 5.40 Å), Phe122 (π-alkyl 4.65, 4.44, 5.40 and 4.76 Å), Tyr209 (π-alkyl 4.87 Å), Trp219 (π-alkyl 5.06 Å), Cys298 (Alkyl 5.39 Å)
Protopanaxatriol	−6.39	Val47 (2.37 Å), Gln49 (2.48 Å)	Trp20 (π-alkyl 4.31 and 4.56 Å), Lys21 (Alkyl 5.20 Å), Pro24 (Alkyl 5.18 Å), Val47 (Alkyl 3.83 Å), Trp79 (π-alkyl 4.97 and 4.60 Å), Trp111 (π-alkyl 5.10, 5.36, 5.10 and 4.55 Å), Phe122 (π-alkyl 5.21, 4.49 and 4.54 Å), Pro218 (Alkyl 4.28 and 3.68 Å), Cys298 (Alkyl 4.45 Å)
Quercetin	−21.86	Gln183 (2.04 Å), Asp216 (1.69 Å), Ile260 (2.22 Å), Lys262 (3.12 and 2.92 Å)	Trp20 (π-π T shaped 5.38 Å), Tyr48 (π-π T shaped 5.50 Å and π-donor 3.12 Å), Tyr209 (π-π stacked 4.36 Å), Ser210 (π-donor 3.94 Å), Lys262 (π-alkyl 4.24 and 4.87 Å), Cys298 (π-sulfur 5.65 Å)
Zenarestat	−31.35	Thr19 (2.69 Å), Trp20 (2.53 Å) Lys77 (2.72 Å), Ser159 (3.45 Å) Asn160 (3.04 Å), Cys298 (3.49 Å)	Trp20 (π-π T shaped 5.79, 5.32 and 5.78 Å, π-sigma 3.48 Å), Tyr48 (π-π T shaped 5.83 Å, and π-donor 3.97 and 3.49 Å), His110 (Fluorine 3.42 and 3.35 Å), Tyr209 (π-π stacked 4.29 Å), Cys298 (π-sulfur 5.85 and 5.94 Å)

**Table 4 molecules-27-02134-t004:** Inhibitory effect of rat lens aldose reductase-active ginsenoside derivatives on sorbitol accumulation in rat lens.

Compounds	Sorbitol Content (mg)/Lens Wet Weight (g)	Inhibition (%)
Blank (glucose-free)	-	-
Control	1.72 ± 0.03	-
Protopanaxadiol	0.06 ± 0.01 ^a^	96.51 ± 2.91 ^a^
(20*S*) ginsenoside Rg3	0.37 ± 0.01 ^b^	78.48 ± 4.27 ^b^
(20*R*) ginsenoside Rg3	0.89 ± 0.01 ^c^	48.25 ± 3.99 ^c^
Ginsenoside Rh2	0.05 ± 0.01 ^a^	97.09 ± 5.41 ^a^
Compound K	0.46 ± 0.01 ^b^	73.25 ± 8.11 ^b^
Protopanaxatriol	0.09 ± 0.01 ^a^	94.76 ± 3.46 ^a^
Ginsenoside Rf	0.28 ± 0.01 ^a^	83.72 ± 4.88 ^a^
(20*S*) ginsenoside Rg2	0.73 ± 0.01 ^c^	57.55 ± 5.11 ^c^
(20*R*) ginsenoside Rg2	0.83 ± 0.01 ^c^	51.74 ± 2.99 ^c^
Ginsenoside Rh1	0.10 ± 0.01 ^a^	94.18 ± 4.71 ^a^
Quercetin ^a^	0.22 ± 0.01 ^a^	87.20 ± 3.91 ^a^

Results are presented as mean ± SD (*n* = 3). Values within a column marked with different letters are significantly different from each other (*p* < 0.05). Samples’ concentration was used at 5 μg/mL on sorbitol accumulation in rat lens. ^a^ Quercetin was used as the positive control.

## Data Availability

Not applicable.

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
