# Peer review of "Inhibition of Aldose Reductase by Ginsenoside Derivatives via a Specific Structure Activity Relationship with Kinetics Mechanism and Molecular Docking Study"

_molecules, 2022, doi:10.3390/molecules27072134_

Round 1
Reviewer 1 Report
The authors aimed to evaluate the rat lens (RLAR; PDB: 3RX2)/recombinant human (HRAR; PDB: 1IEI) aldolase (involved in diabetic retinopathy) inhibitory activity of twenty-two purified ginsenosides using a QSAR-based approach [Enzyme inhibition in vitro/ex vivo + molecular docking simulation (LeadIT software/ HYDE assessment). The scientific soundness and uniqueness of the study lay on the systematic evaluation of many naturally-occurring ginsenosides to draw conclusions on specific molecular moieties involved in aldose inhibition, so new and more effective molecules could be synthesized. To improve the manuscript´s scientific contribution, authors are asked to consider the following:
General
- The readability and syntax of the manuscript will be substantially improved if it is reviewed by a formal translation agency or by a native English spoken person.
Sections
- Title. Quite long. Suggestion: Anti-aldolase activity of ginsenosides: A comprehensive Quantitative structure-activity relationship evaluation.
- Abstract. OK.
- Introduction. Too long. Authors should focus on supporting the lack of systematic information on ginsenosides´ anti-aldolase activity and particularly of QSAR-based evaluations on this subject.
- Results/discussion. OK.
- Methods. Section 4.2: OK.
- Tables. Should be formatted according to Molecules´ guidelines.
- Figures. Improve the resolution of all figures (≥300 dpi).
- Conclusion. Change if necessary, according to the new suggestions
- References. Authors must reduce the number of references≥ 10y old to say 20% or less and references’ format should comply with Molecules´ guidelines.
Author Response
Reviewer 1#
Query: The authors aimed to evaluate the rat lens (RLAR; PDB: 3RX2)/recombinant human (HRAR; PDB: 1IEI) aldolase (involved in diabetic retinopathy) inhibitory activity of twenty-two purified ginsenosides using a QSAR-based approach [Enzyme inhibition in vitro/ex vivo + molecular docking simulation (LeadIT software/ HYDE assessment). The scientific soundness and uniqueness of the study lay on the systematic evaluation of many naturally occurring ginsenosides to draw conclusions on specific molecular moieties involved in aldose inhibition, so new and more effective molecules could be synthesized. To improve the manuscript´s scientific contribution, authors are asked to consider the following.
Response: Thank you very much for your valuable comments regarding our manuscript.
General Query: The readability and syntax of the manuscript will be substantially improved if it is reviewed by a formal translation agency or by a native English spoken person.
Response: Thank you very much for your kind suggestions and valuable comments regarding our manuscript. According to the reviewer's suggestions, we revised our whole manuscript and edited it by English native speaker. We submitted an English editing certificate during our submission.
Query: Title. Quite long. Suggestion: Anti-aldolase activity of ginsenosides: A comprehensive Quantitative structure-activity relationship evaluation.
Response: Thank you very much for your valuable suggestions regarding our manuscript. According to the other two reviewer’s suggestions, we will retain the same title of the manuscript, but we greatly appreciate the reviewer's suggestion.
Query: Abstract. OK.
Response: Thank you very much for your valuable comments regarding our manuscript.
Query: Too long. Authors should focus on supporting the lack of systematic information on ginsenosides´ anti-aldolase activity and particularly of QSAR-based evaluations on this subject.
Response: Thank you very much for your kind suggestions and valuable comments regarding our manuscript. According to the reviewer's suggestions, we have revised the introduction section and marked it as red color.
Query: Results/discussion. OK.
Response: Thank you very much for your valuable comments regarding our manuscript.
Query: Methods. Section 4.2: OK.
Response: Thank you very much for your valuable comments regarding our manuscript.
Query: Tables. Should be formatted according to Molecules´ guidelines.
Response: We have formatted according to the Molecules guideline.
Query: Figures. Improve the resolution of all figures (≥300 dpi).
Response: Thank you very much for your valuable comments regarding our manuscript. We have improved figures, please see the improve figure throughout the manuscript.
Query: Conclusion. Change if necessary, according to the new suggestions.
Response: Thank you very much for your kind suggestions and valuable comments regarding our manuscript.
Query: References. Authors must reduce the number of references≥ 10y old to say 20% or less and references’ format should comply with Molecules´ guidelines.
Response: We have followed the Molecules guideline.
Reviewer 2 Report
Reviewer report:
Dear authors,
I completed the revision of manuscript entitled "Inhibition of aldose reductase by ginsenoside derivatives via a 2 specific structure activity relationship with kinetics mechanism 3 and molecular docking study"
Below you can find my comments
This study deals to evaluate the anti-diabetic potential of 22 ginsenosides via inhibition against rat lens aldose reductase (RLAR), and human recombinant aldose reductase (HRAR), an enzyme implicated in the polyol pathway and responsible for the onset of long-term diabetic complications.
In addition, authors have previously reported that ginsenosides possess promising antiglycation as well as antioxidant properties that may be able to prevent the progression of diabetic complications. Authors conducted extensive SAR studies on the rat lens as well as human recombinant aldose reductase inhibitory activities of several ginsenosides derivatives. They have found that four ginsenosides exhibited good RLAR inhibitory activities on micromolar range (0.67 - 7.28 μM), while another four ginsenosides exhibited inhibitory activities against HRAR with IC50 values in the range of 0.36 - 4.66 μM.
The authors also performed enzyme kinetic analyses of ginsenosides to confirm the type of enzymatic inhibition. The interactions between these ginsenosides and RLAR and HRAR were simulated using extensive molecular docking analysis, and their docking energies as well as mechanisms of enzymes’ inhibition were examined. Finally, authors investigated the effect of active ginsenosides on sorbitol accumulation to explore their potential to treat diabetic complications. Specifically, authors found that four ginsenosides (Protopanaxadiol, Ginsenoside Rh2, Protopanaxatriol, and Ginsenoside Rh1) demonstrated over 94% inhibitory effect of aldose reductase-active ginsenosides on sorbitol accumulation in rat lens.
Overall, I find this study interesting. The manuscript is well written. The introductory section explains in depth the subject of this article, while the experimental section is well organized. The only bias I observed concern the in vitro pharmacological assessments. The authors only showed that ginsenosides inhibit the rat and human aldose reductase enzymes with moderate to good IC50 values. However, a great number of literature data exhibited that besides aldose reductase, ARIs may also interfere with the inhibition of aldehyde reductase (ALR1). ALR1 is another member of the aldoketoreductase superfamily being closely related to aldose reductase and specifically responsible for the detoxification of harmful aldehydes. Therefore, I think it would be advisable for the authors to perform in vitro tests to evaluate the inhibition activities of the most promising ginsenosides against aldehyde reductase. The selectivity profiles of compounds could strengthen the in vitro results and clarify the selection criteria of these compounds for further in vivo studies.
Author Response
Reviewer 2#
General Query: This study deals to evaluate the anti-diabetic potential of 22 ginsenosides via inhibition against rat lens aldose reductase (RLAR), and human recombinant aldose reductase (HRAR), an enzyme implicated in the polyol pathway and responsible for the onset of long-term diabetic complications. In addition, authors have previously reported that ginsenosides possess promising antiglycation as well as antioxidant properties that may be able to prevent the progression of diabetic complications. Authors conducted extensive SAR studies on the rat lens as well as human recombinant aldose reductase inhibitory activities of several ginsenosides derivatives. They have found that four ginsenosides exhibited good RLAR inhibitory activities on micromolar range (0.67-7.28 μM), while another four ginsenosides exhibited inhibitory activities against HRAR with IC50 values in the range of 0.36-4.66 μM. The authors also performed enzyme kinetic analyses of ginsenosides to confirm the type of enzymatic inhibition. The interactions between these ginsenosides and RLAR and HRAR were simulated using extensive molecular docking analysis, and their docking energies as well as mechanisms of enzymes’ inhibition were examined. Finally, authors investigated the effect of active ginsenosides on sorbitol accumulation to explore their potential to treat diabetic complications. Specifically, authors found that four ginsenosides (Protopanaxadiol, Ginsenoside Rh2, Protopanaxatriol, and Ginsenoside Rh1) demonstrated over 94% inhibitory effect of aldose reductase-active ginsenosides on sorbitol accumulation in rat lens.
Response: Thank you very much for your valuable comments regarding our manuscript.
Query: Overall, I find this study interesting. The manuscript is well written. The introductory section explains in depth the subject of this article, while the experimental section is well organized. The only bias I observed concern the in vitro pharmacological assessments. The authors only showed that ginsenosides inhibit the rat and human aldose reductase enzymes with moderate to good IC50 values. However, a great number of literature data exhibited that besides aldose reductase, ARIs may also interfere with the inhibition of aldehyde reductase (ALR1). ALR1 is another member of the aldoketoreductase superfamily being closely related to aldose reductase and specifically responsible for the detoxification of harmful aldehydes. Therefore, I think it would be advisable for the authors to perform in vitro tests to evaluate the inhibition activities of the most promising ginsenosides against aldehyde reductase. The selectivity profiles of compounds could strengthen the in vitro results and clarify the selection criteria of these compounds for further in vivo studies.
Response: Thank you very much for your valuable comments regarding our manuscript. We are agreeing with the reviewer's opinions that others reductase families like ALR1, ALR2, and ALR3 also have important diabetic complications, but the physiological role of ALR1 and ALR3 has not yet been well identified for pathogenesis of diabetic complications. However, aldose reductase (ALR2) that could reduce excess glucose to sorbitol in diabetes mellitus has implicated the enzyme in the pathogenesis of diabetic complications affecting the eyes, kidneys, and nervous system. Therefore, we focused only on rat and human aldose reductase enzymes. Additionally, the in vivo experiments are underway in our laboratory to clarify their beneficial/harmful effects of these ginsenosides. We hope it successfully addressed the main objective of the manuscript which will provide some necessary information for other interested researchers for further investigation.
Reviewer 3 Report
In the manuscript entitled “Inhibition of aldose reductase by ginsenoside derivatives via a specific structure activity relationship with kinetics mechanism and molecular docking study”, Authors described the inhibitory potential of 22 ginsenosides compounds on NADHPH reductase activity thorough kinetic studies, molecular docking simulation and sorbitol accumulation suppression in rat lenses under high-glucose conditions. The topic would be of interest and worth to be furthered, however, manuscript presents some critical issues regarding the results presented. List of criticisms are described below:
Major critical points:
Authors use a rat lens homogenate to evaluate the inhibitory effect of ginsenoside compounds on aldose reductase enzyme. This correlation could be accurate if is proven that the NADPH dependent D,L-glyceraldehyde reduction is due only to AR activity.
The fact that the sorbitol accumulation in lens culture is reduced in the presence of ginsenoside compounds could indicate that also rat lens Aldose reductase was inhibited, however the IC50 and kinetic constants values evaluated using a rat lens homogenate cannot exclude that the observed inhibitory kinetic effect could be attributable to a mixture of enzymes having aldehyde reductase activity rather than the only aldose reductase enzyme involved in the polyol pathway. In addition, using a crude homogenate cannot exclude the presence of other compounds capable of modify (in term of competition or compounds interactivity) the interaction between the enzyme and the compounds tested (so IC50 values and Ki/ki’ values could be different).
Another point of criticism regards the enzyme kinetic analysis:
-In the case of mixed type inhibitors, both ki and ki’ have to been evaluated through Dixon plots analysis (see e.g. Cornish-Bowden, 1973).
-The x axis values in the double reciprocal plot are not correct.
-For some compounds analysis, the double reciprocal plot control lines intercepts are quite different (e.g. graph i compared to graph m of figure 3) indicating different Km and Vmax values for the enzyme without inhibitor (control kinetic parameters have to be as best as possible the same).
-For some compounds analysis, Ki values listed in table 1 are not matching with the Ki values obtained by the intercepts lines (graph e, f and h in figure 3).
-three substrate and three inhibitor concentration could be too few to obtain a correct interpolation of the lines in the double reciprocal plots and Dixon plots. How lines intercept with the y axis or x axis in the double reciprocal plots are determined? How the ki values are determined in the Dixon plots. Did Authors have strong reliable R2 values to correct define the lines intercept?
Minor points:
Lines 156-157: “Both (20S) Rg3 and (20S) Rg2 showed better RLAR inhibitory activity than (20R) Rg3 and (20R) Rg2 did because they aligned better with the hydroxyl acceptor group.” Did authors declare that on the basis of which speculation?
Lines 233-239: it is not clear that Authors refer to HRAR.
Line 393-395: “Thus, our previous and current results suggest that ginsenosides may possess good inhibitory activities against diabetic complications, as suggested by their significant antioxidant activity.” This declaration is too strong compared to the data obtained.
Section 3.3.: there are no indication about the HRAR enzyme units used in the inhibition assay.
Line 516: the sentence is not clear.
Author Response
Reviewer 3#
General Query: In the manuscript entitled “Inhibition of aldose reductase by ginsenoside derivatives via a specific structure activity relationship with kinetics mechanism and molecular docking study”, Authors described the inhibitory potential of 22 ginsenosides compounds on NADHPH reductase activity thorough kinetic studies, molecular docking simulation and sorbitol accumulation suppression in rat lenses under high-glucose conditions. The topic would be of interest and worth to be furthered, however, manuscript presents some critical issues regarding the results presented. List of criticisms are described below:
Response: We greatly appreciate reviewer 3’s critical comments and suggestions.
Query: Authors use a rat lens homogenate to evaluate the inhibitory effect of ginsenoside compounds on aldose reductase enzyme. This correlation could be accurate if is proven that the NADPH dependent DL-Glyceraldehyde reduction is due only to AR activity.
Response: Thank you very much for your kind suggestions and valuable comments regarding our manuscript. It has been already reported that the correlation of NADPH dependent DL-Glyceraldehyde reduction to AR activity (J.Biol.Chem.1992,267:29,20965-20970; Br.J. Ophthalmol. 1983,67,696-699; J.Biol.Chem. 1965, 240:2, 882-887)
Query: The fact that the sorbitol accumulation in lens culture is reduced in the presence of ginsenoside compounds could indicate that also rat lens Aldose reductase was inhibited, however the IC50 and kinetic constants values evaluated using a rat lens homogenate cannot exclude that the observed inhibitory kinetic effect could be attributable to a mixture of enzymes having aldehyde reductase activity rather than the only aldose reductase enzyme involved in the polyol pathway. In addition, using a crude homogenate cannot exclude the presence of other compounds capable of modify (in term of competition or compounds interactivity) the interaction between the enzyme and the compounds tested (so IC50 values and Ki/Ki’ values could be different).
Response: We greatly appreciate reviewer 3’s critical comments. Our laboratory has an established facility and protocols for isolation and purification of rat lens homogenate.
Query: -In the case of mixed type inhibitors, both Ki and Ki’ have to been evaluated through Dixon plots analysis (see e.g Cornish-Bowden, 1973).
Response: Thank you very much for your comments regarding our manuscript. According to the reviewer's suggestion, we have included Kiu (binding constants of inhibitor with enzyme-substrate complex) value for mixed type inhibitors in table 1, marked as red color.
Query: -The x axis values in the double reciprocal plot are not correct.
Response: We greatly appreciate reviewer 3’s comments and fixed it.
Query: For some compounds analysis, the double reciprocal plot control lines intercepts are quite different (e.g graph i compared to graph m of figure 3) indicating different Km and Vmax values for the enzyme without inhibitor (control kinetic parameters have to be as best as possible the same).
Response: Thank you very much for your kind suggestions and valuable comments regarding our manuscript. According to the reviewer suggestions, we have revised figure 3m, please see the revised figure. Additionally, if we used different concentrations of compounds (inhibitors) their double reciprocal plot control lines intercepts (including Km and Vmax values) would be quite different even with the same type of inhibition.
Query: For some compounds analysis, Ki values listed in table 1 are not matching with the Ki values obtained by the intercepts lines (graph e, f and h in figure 3).
Response: Thank you very much for your valuable comments regarding our manuscript. According to the reviewer suggestion, we have revised the Ki values in Table 1, marked as red color.
Query: Three substrate and three inhibitor concentrations could be too few to obtain a correct interpolation of the lines in the double reciprocal plots and Dixon plots. How lines intercept with the y axis or x axis in the double reciprocal plots are determined? How the Ki values are determined in the Dixon plots. Did Authors have strong reliable R2 values to correct define the lines intercept?
Response: Thank you very much for your valuable comments regarding our manuscript. As some compounds were not so strong inhibitors against RLAR and HRAR, therefore, we did not use more than three different concentrations of substrate and inhibitors. Primarily, we used several concentrations of substrate and inhibitors (selected lowest and highest concentrations), but it did not change the curve fitting (with reliable R2 values), and Ki value, as well as inhibition types. We hope it successfully addressed the main objective of the manuscript which will provide some necessary information for other interested researchers for further investigation.
Query: Lines 156-157: “Both (20S) Rg3 and (20S) Rg2 showed better RLAR inhibitory activity than (20R) Rg3 and (20R) Rg2 did because they aligned better with the hydroxyl acceptor group.” Did authors declare that on the basis of which speculation?
Response: Thank you very much for your valuable comments regarding our manuscript. 20(S) and 20(R) are stereoisomers of each other that depend on the position of the C-20 hydroxyl in ginsenosides. 20(S)-OH is geometrically close to the C-12 hydroxyl of ginsenosides. 20(R)-OH is far from the C-12 hydroxyl. Both (20S) Rg3 and (20S) Rg2 showed better RLAR inhibitory activity than (20R) Rg3 and (20R) Rg2 did because they aligned better with the hydroxyl acceptor group.
Query: Lines 233-239: it is not clear that Authors refer to HRAR.
Response: Thank you very much for your valuable comments regarding our manuscript. In lines 233-239, we explained mixed and non-competitive types of inhibition kinetics mechanism, by showing Km, and Vmax condition.
Query: Line 393-395: “Thus, our previous and current results suggest that ginsenosides may possess good inhibitory activities against diabetic complications, as suggested by their significant antioxidant activity.” This declaration is too strong compared to the data obtained.
Response: We greatly appreciate reviewer 3’s comments. But it has been already reported that ginsenosides have remarkable pharmacological activity not only antioxidants and diabetes but also wide ranges of biological activity.
Query: Section 3.3.: there are no indication about the HRAR enzyme units used in the inhibition assay.
Response: Thank you very much for your valuable comments regarding our manuscript. For HRAR inhibition, 0.4 U HRAR enzyme units were used for inhibition evaluations. We have included this information in section 3.3 marked as red color.
Query: Line 516: the sentence is not clear.
Response: Thank you very much for your valuable comments regarding our manuscript. We have concise this sentence and marked as red color in line 516.
Round 2
Reviewer 2 Report
After reading the revised manuscript, I think that it can be accepted.
Author Response
Reviewers 2#
Query: After reading the revised manuscript, I think that it can be accepted.
Response: We greatly appreciate reviewer comments.
Reviewer 3 Report
Some point of criticism still remain.
Query: Authors use a rat lens homogenate to evaluate the inhibitory effect of ginsenoside compounds on aldose reductase enzyme. This correlation could be accurate if is proven that the NADPH dependent DL-Glyceraldehyde reduction is due only to AR activity.
Response: Thank you very much for your kind suggestions and valuable comments regarding our manuscript. It has been already reported that the correlation of NADPH dependent DL-Glyceraldehyde reduction to AR activity (J.Biol.Chem.1992,267:29,20965-20970; Br.J. Ophthalmol. 1983,67,696-699; J.Biol.Chem. 1965, 240:2, 882-887)
Review response: the NADPH dependent AR activity on DL-Glyceraldehyde reduction is out of debate; my criticism concerns using a homogenate extract which could be contain other NADPH dependent reductase activity enzymes as well as AR. The reference J.Biol.Chem.1992,267:29,20965-20970 reports an enzymatic characterization of the human aldose reductase recombinant which has been purified; the reference Br.J. Ophthalmol. 1983,67,696-699 reports AR aldose reductase activity in human lens and the enzyme assay protocol is referred to an article of Crabbe and Halder 1979 in which the bovine lens aldose reductase purification procedure has been described. Anyway, the focus of the problem regards the reliability of using an entire homogenate extract to assay the catalytic action of a specific enzyme.
Query: The fact that the sorbitol accumulation in lens culture is reduced in the presence of ginsenoside compounds could indicate that also rat lens Aldose reductase was inhibited, however the IC50 and kinetic constants values evaluated using a rat lens homogenate cannot exclude that the observed inhibitory kinetic effect could be attributable to a mixture of enzymes having aldehyde reductase activity rather than the only aldose reductase enzyme involved in the polyol pathway. In addition, using a crude homogenate cannot exclude the presence of other compounds capable of modify (in term of competition or compounds interactivity) the interaction between the enzyme and the compounds tested (so IC50 values and Ki/Ki’ values could be different).
Response: We greatly appreciate reviewer 3’s critical comments. Our laboratory has an established facility and protocols for isolation and purification of rat lens homogenate.
Review response: again the point of my criticism regards using a rat lens homogenate (which is purified by what?) for evaluating an AR specificity activity instead of using the AR purified enzyme. HRAR and RLAR are not the same in term of purity grade. Authors have evaluated a NADPH reductase activity. In the presence of ginsenoside compounds, sorbitol accumulation is inhibited so AR activity is inhibited but in term of IC50 and Ki/ki’ values, these data have to been referred to a NADPH reductase activity using DL-Glyceraldehyde as substrate.
Query: -The x axis values in the double reciprocal plot are not correct.
Response: We greatly appreciate reviewer 3’s comments and fixed it.
Review response: in the revised manuscript the x axis of figure 2 and 3 have not been fixed; the substrate concentration has to be 1/[S] but are reported as [S].
Query: For some compounds analysis, the double reciprocal plot control lines intercepts are quite different (e.g graph i compared to graph m of figure 3) indicating different Km and Vmax values for the enzyme without inhibitor (control kinetic parameters have to be as best as possible the same).
Response: Thank you very much for your kind suggestions and valuable comments regarding our manuscript. According to the reviewer suggestions, we have revised figure 3m, please see the revised figure. Additionally, if we used different concentrations of compounds (inhibitors) their double reciprocal plot control lines intercepts (including Km and Vmax values) would be quite different even with the same type of inhibition.
Review response: control values obtained have not to be dependent by the different concentrations of compounds (inhibitors) used. Has 3m figure been revised only in terms of layout or also in terms of experimental values obtained?
Author Response
Reviewer 3#
Reviewer Query:
The NADPH dependent AR activity on DL-Glyceraldehyde reduction is out of debate; my criticism concerns using a homogenate extract which could be contain other NADPH dependent reductase activity enzymes as well as AR. The reference J.Biol.Chem.1992,267:29,20965-20970 reports an enzymatic characterization of the human aldose reductase recombinant which has been purified; the reference Br.J. Ophthalmol. 1983,67,696-699 reports AR aldose reductase activity in human lens and the enzyme assay protocol is referred to an article of Crabbe and Halder 1979 in which the bovine lens aldose reductase purification procedure has been described. Anyway, the focus of the problem regards the reliability of using an entire homogenate extract to assay the catalytic action of a specific enzyme.
Response: We greatly appreciate reviewer comments. We agree with the reviewer opinion that using the whole rat lens homogenate extract is less specific, but most of the researchers are using rat lens homogenate extract along with human aldose reductase recombinant enzyme (more reliable) and finally do in vivo experiments. We are also in the same tract, we did RLAR and HRAR and currently our laboratory testing on several ginsenoside derivatives in vivo models. We hope it successfully addressed the main objective of the manuscript which will provide some necessary information for other interested researchers for further investigation.
Review Query: The NADPH dependent AR activity on DL-Glyceraldehyde reduction is out of debate; Again, the point of my criticism regards using a rat lens homogenate (which is purified by what?) for evaluating an AR specificity activity instead of using the AR purified enzyme. HRAR and RLAR are not the same in term of purity grade. Authors have evaluated a NADPH reductase activity. In the presence of ginsenoside compounds, sorbitol accumulation is inhibited so AR activity is inhibited but in term of IC50 and Ki/ki’ values, these data have to been referred to a NADPH reductase activity using DL-Glyceraldehyde as substrate.
Response: We greatly appreciate reviewer comments. We agree with the reviewer opinion that using the whole rat lens homogenate extract is less specific, but most of the researchers are using rat lens homogenate extract along with human aldose reductase recombinant enzyme (more reliable) and finally do in vivo experiments. We are also in the same tract, we did RLAR and HRAR and currently in vivo experiments are going on in our laboratory. We hope it successfully addressed the main objective of the manuscript which will provide some necessary information for other interested researchers for further investigation.
Review Query: in the revised manuscript the x axis of figure 2 and 3 have not been fixed; the substrate concentration has to be 1/[S] but are reported as [S].
Response: We greatly appreciate reviewer comments. We have revised the figures 2 and 3 and corrected 1/[S] instead of [S]. Please see the revised figures 2i-p, and figure 3i-p.
Review Query: control values obtained have not to be dependent by the different concentrations of compounds (inhibitors) used. Has 3m figure been revised only in terms of layout or also in terms of experimental values obtained?
Response: We greatly appreciate reviewer comments. We have corrected figures 2i-p, and figure 3i-p.